# A Novel Enzootic Nasal Tumor Virus Circulating in Goats from Southern China

**DOI:** 10.3390/v11100956

**Published:** 2019-10-17

**Authors:** Shao-Lun Zhai, Dian-Hong Lv, Zhi-Hong Xu, Jie-Shi Yu, Xiao-Hui Wen, He Zhang, Qin-Ling Chen, Chun-Ling Jia, Xiu-Rong Zhou, Qi Zhai, Feng Li, Patrick C. Y. Woo, Susanna K. P. Lau, Dan Wang, Wen-Kang Wei

**Affiliations:** 1Animal Disease Diagnostic Center, Institute of Animal Health, Guangdong Academy of Agricultural Sciences, Scientific Observation and Experiment Station of Veterinary Drugs and Diagnostic Techniques of Guangdong Province, Ministry of Agriculture and Rural Affairs, Key Laboratory of Animal Disease Prevention of Guangdong province, Guangzhou 510640, China; zhaishaolun@163.com (S.-L.Z.); lvdianhong@gdaas.cn (D.-H.L.); xuzhihong@gdaas.cn (Z.-H.X.); wenxiaohui@gdaas.cn (X.-H.W.); chenqinling@gdaas.cn (Q.-L.C.); jiachunling@gdaas.cn (C.-L.J.);; 2Department of Biology and Microbiology, South Dakota State University, Brookings, SD 57007, USA; Jieshi.yu@sdstate.edu (J.-S.Y.); Feng.Li@sdstate.edu (F.L.); 3State Key Laboratory of Emerging Infectious Diseases, The University of Hong Kong, Hong Kong, China; pcywoo@hku.hk; 4Department of Microbiology, The University of Hong Kong, Hong Kong, China; 5Collaborative Innovation Center for Diagnosis and Treatment of Infectious Diseases, The University of Hong Kong, Hong Kong, China

**Keywords:** enzootic nasal tumor virus, ENTV-2, novel variation, genetic diversity, goats, Guangdong province

## Abstract

Enzootic nasal tumor virus (ENTV) has two types, ENTV-1 in sheep and ENTV-2 in goats, respectively. In China, the incidence of ENTV-2 related diseases has increased year by year. In this study, we reported an outbreak of ENTV-2 in a commercial goat farm in Qingyuan city, Guangdong province, southern China. A full-length genome of ENTV-2 (designated GDQY2017), with 7479 base pairs, was sequenced. Although GDQY2017 shared the highest nucleotide identity with a Chinese ENTV-2 isolate (ENTV-2CHN4, GenBank accession number KU258873), it possesses distinct genome characteristics undescribed, including a non-continuous 21-nucleotide insertion in the *gag* gene and a non-continuous 12-nucleotide deletion in the *env* gene. Notably, most of these indel nucleotide sequences were originated from a Chinese jaagsiekte sheep retrovirus (JSRV) isolate (GenBank accession number DQ838494). In the *gag* and *env* genes, GDQY2017 was phylogenetically related to those Chinese ENTV-2 isolates and a Chinese JSRV isolate (DQ838494). For GDQY2017-like viruses, more surveillance work should be made to explain their pathogenicity in goat herds. To our knowledge, this study represents the first to demonstrate the circulating pattern of ENTV-2 in Guangdong province, China, which will help to better understand the epidemiology and genetic diversity of ENTV-2.

## 1. Introduction

Enzootic nasal tumor virus (ENTV) is a single positive-stranded RNA virus that belongs to the genus *Betaretrovirus* in the family *Retroviridae*. The genome of ENTV is about 7.5 kilo-base pairs and contains four open reading frames (ORFs), which encode *gag* (group antigen) protein, *pro* protease protein, *pol* protein, and *env* protein, respectively. Genetically, ENTV is close to jaagsiekte sheep retrovirus (JSRV, another ovine betaretrovirus), which is the causative agent of ovine pulmonary adenocarcinoma (OPA) [1]. ENTV has two types, ENTV-1 in sheep and ENTV-2 in goats [2,3], which mainly cause enzootic nasal adenocarcinoma (ENA), including copious serous nasal discharge, dyspnea, and even death due to respiratory failure [4,5]. 

Previous studies showed that ENTV-2 was mainly found in Inner Mongolia, Hunan, and Sichuan [6,7,8,9], while in past five years, ENTV-2 has spread to Anhui, Shaanxi, Chongqing, Hunan, and Fujian, suggesting a potential outbreak of ENTV-2 in China [10,11,12,13,14]. In this study, we reported the first case of ENTV-2 infection with distinct genome characteristics in Guangdong, China.

## 2. Materials and Methods 

### 2.1. Case Description

In September 2017, a suspected case of ENA occurred on a goat farm (total livestock: 82) in Qingyuan city, a city of Guangdong province in southern China. The eight affected goats aged from three-month to one-year. The affected goats presented initial symptoms as nasal polyps (Figure 1a), followed by copious serous nasal discharge, dyspnea emerged, and even death (Figure 1b). To define potential causes of ENA, one affected goat (goat 3) was sent to the Animal Disease Diagnostic Center, Institute of Animal Health, Guangdong Academy of Agricultural Sciences. After euthanasia, the autopsy report showed that the main pathological changes (tumor-like tissue hyperplasia) were observed within the goat nose (Figure 1c). These clinical manifestations suggested that ENTV-2 was considered the potential cause. Serum, tumor-like tissue, and nasal discharge of goat 3 were collected and stored at −80 °C for further use. Tumor-like tissues were stained by hematoxylin-eosin (HE). Sample collection was approved by the ethical and ethics commission (Institute of Animal Health, Guangdong Academy of Agricultural Sciences, China). The license number was SYXK(Yue) 2011–0116. Moreover, all methods in this study were performed in accordance with national and local laws and guidelines.

### 2.2. Detection and Complete Genome Amplification of ENTV-2

One-step RT-PCR kit (Takara Inc., Dalian, Liaoning, China) was performed to detect ENTV-2 in collected samples, using a pair of previously reported primers (3F: 5′-CACTCCTAATTTGTGCCCACG-3′ and 3R: 5′-GGCCACTGATCGACCCATAC-3′) targeting gag-pro-pol fusion gene (Table 1), according to the manufacturer’s instructions. Four additional pairs of primers (Table 1) were synthesized and used to amplify the genome of ENTV-2, according to one previous literature [13]. One step RT-PCRs were performed with a final reaction volume of 50 μL, containing 25 μL 2 × RT-PCR buffer (Takara, Inc., Dalian, Liaoning, China), 1 μL Enzyme Mix (including reverse transcriptase and DNA polymerase), 5 μL viral RNA, 1 μL each of the primers (10 pmol), and 17 μL RNase-free water. Cycling conditions were as follows: 50 °C for 30 min, 94 °C for 3 min followed by 30 cycles of 94 °C for 30 s, 55 °C for 30 s, and 72 °C for 70 s. A final extension of 5 min at 72 °C concluded the program. The PCR products were purified and cloned into the pGEM-T vector (Tiangen Biotech Co., Ltd. Beijing, China). In addition, the positive recombinant plasmids were sequenced by the Sanger sequencing (Sangon Biotech Co., Ltd. Shanghai, China).

### 2.3. Sequence and Phylogenetic Analysis

Nucleotide sequence editing, full-length genome assembling, amino acid sequence prediction, sequence alignment, and analysis were performed by using Lasergene v7.1 sequence analysis software package (DNASTAR Inc., Madison, WI, USA). Phylogenetic analysis was performed by the maximum likelihood method with 1000 bootstrap replicates using MEGA 7 software (https://www.megasoftware.net/). One thousand bootstrap replicates were used. Recombination analysis was performed using a recombination detection program (RDP) 4.95 (http://web.cbio.uct.ac.za/~darren/rdp.html). 

## 3. Results and Discussion

According to the results of HE staining (Figure 2), the tumor appeared in the goat was considered to be nasal adenoma, which was in compliance with the feature of ENA.

Serum, tumor-like tissue, and nasal discharge of goat 3 were positive for ENTV-2 by RT-PCR detection. A complete genome sequence (designated GDQY2017) of ENTV-2 with 7479 base pairs was obtained from the goat sera and deposited in the GenBank database (accession number: MK164396). GDQY2017 has four main encoding gene regions, *gag* gene (position 266–2125), *pro* gene (position 2017–2886), *pol* gene (position 3132–4259), *env* gene (position 5351–7207). Complete genome alignment results showed that GDQY2017 had the highest nucleotide similarity (96.6%) with a Chinese ENTV-2 strain (ENTV-2CHN4, GenBank accession number KU258873), and the lowest nucleotide similarity (88.2%) with a Spanish ENTV-2 strain (GenBank accession number AY197548) isolated in the 1990s (Table 2). Compared with ENTV-1, JSRV, endogenous JSRV (enJSRV), and other ENTV-2 sequences, the *gag* gene and *env* gene of GDQY2017 had higher diversities (85.7–96.8% vs. 85.8–95%) than the *pro* gene and *pol* gene (90.9–99.4% vs. 88–99.2%). 

In addition to nucleotide substitution, the *gag* gene of GDQY2017 had a distinctive non-continuous 21-nucleotide insertion (Figure 3). Another difference from most of the ENTV-2 sequences is a non-continuous 12-nucleotide deletion in the *env* gene (Figure 4). Interestingly, most of these indel nucleotide sequences were originated from a JSRV isolate (DQ838494), and some nucleotide sequences were identical to one endogenous JSRV isolate (enJSRV-1, GenBank accession number DQ838494) (Figure 3 and Figure 4).

At the amino acid level, the gag protein sequences of GDQY2017 showed the highest similarity (95.1%) with ENTV-2CHN6 (KU258875), ENTV-2CHN7 (KU258876), and ENTV-2CHN11 (KU258880), and shared the lowest similarity (88.3%) with CQ1 (MK164400) and ENTV-SC (HM104174). Compared to previous strains, there were seven non-continuous amino acid insertion sites in the gag protein of GDQY2017 (Figure 5). While, the env protein sequences of GDQY2017 had the highest similarity (94.6%) with one endogenous JSRV isolate (KarM, MF175071), and shared the lowest similarity with a Spanish ENTV-2 isolate (AY197548). Compared to previous ENTV-2 strains, four continuous amino acid deletion sites were found in the env protein of GDQY2017 (Figure 6), which is identical to ENTV-2CHN1 (KU258870). 

In the phylogenetic tree based on the *gag* and *env* genes, GDQY2017 was clustered with ENTV-2 branch and phylogenetically related to a Chinese JSRV isolate (DQ838494) or an endogenous JSRV isolate (enJSRV-1, EF680311) (Figure 7 and Figure 8).

Moreover, no possible recombination points among ENTV-2 and other retroviruses were found by RDP. 

In China, reports of OPA and ENA could be traced back to 1982 in Inner Mongolia of north China [6]. Although cases of OPA and ENA were restricted in northern China previously, increasing cases have been reported in recent years [6,7]. ENTV spread from northern to southern gradually, including Sichuan province, Chongqing city, Guizhou province, Fujian province, Anhui province, and Hunan province [8,9,10,11,12,13,14]. Here, we reported the first ENTV-2 case in Guangdong province, southern China. The possible reasons for this outbreak are as follows: 1) the rapid development of the logistics industry and the policy guidance for southern herbivore animal husbandry that made that live goats move around more and more frequently; 2) the lack of ENTV-2 pathogen quarantine when farms introduce or export goats. Therefore, effective animal disease surveillance agencies in China still need improvement.

One previous study showed that an unusually high degree of genetic stability (>96%) existed in the genome of ENTV-1 isolated from North America and Europe [15]. However, JSRV isolates had lower genetic similarity (89.9–95.7%) than ENTV-1 isolates, showing three main variable regions (VRs) (VR1 and VR2 in *gag* genes, and VR3 in *env* gene) [16,17,18]. In this study, the genome similarity of ENTV-2 varied from 87.1% to 99.8%, indicating that ENTV-2 had higher genetic diversity than ENTV-1 and JSRV. Notably, we identified a novel ENTV-2 (GDQY2017) with gene insertion and deletion among the genome sequence that is significantly different from previously reported ENTV-2 strains. Although GDQY2017 is similar to the ENTV-2CHN1 and the Chinese JSRV sequence (DQ838494) and enJSRV sequence, it had the same six-amino acid deletion in the *env* gene, and ENTV-2CHN1 had an additional amino acid (N) insertion in the *env* gene (Figure 6). Other than previous ENTV-2 sequences, GDQY2017 had a higher identity (92.2%) to a Chinese JSRV sequence (DQ838494) and an endogenous JSRV sequence (enJSRV-1, EF680311) than other JSRV and ENTV-1 sequences (87.6–88.4%). Interestingly, some sequences of GDQY2017 were similar or identical to certain sequences of the Chinese JSRV sequence (DQ838494) and enJSRV-1 (EF680311) (Figure 3 and Figure 4). Surprisingly, most of the indel points in the GDQY2017 genome located in VR regions of the JSRV genome [16,17,18]. All the genome features above indicated that GDQY2017 was a novel ENTV-2 isolate with unique features among JSRV and enJSRV.

In this study, five pairs of overlapping primers were used to amplify the genome of ENTV-2 and derived five PCR products were cloned and sequenced. Due to high sequence similarity between both exogenous and endogenous viruses, it is still possible that some of determined sequences may come from endogenous viral sequences. Further experiment is needed towards amplifying and sequencing the full genome-size clone or large PCR products so any possible components of the endogenous viruses would be excluded. 

Generally, ENTV-1 and JSRV only infected sheep, while ENTV-2 only infected goats. But, one previous study reported that the experimental goats could be infected with JSRV and developed lung tumors [19], which improve our understanding of retrovirus in sheep and goats, and revealed that JSRV could break across the species barriers. In China, goats carrying ENTV-2 and sheep carrying JSRV were raised together on some farms. The feeding mode might potentially promote the emergence of the chimeric virus. In this study, GDQY2017 had certain identical or similar gene sequences with the Chinese JSRV sequence (DQ838494), which further supports our speculation. As far as we checked, any clinical lesions in the lungs of the affected goats were not observed (data not shown). In subsequent studies, relevant animal experiments should be made to verify this hypothesis.

Usually, ENTV-2 infections occur in goats older than six-month-old [6,20,21]. However, in this study, we found that younger goats (about 3-month old) could be also infected with ENTV-2 and showed tumor-like lesions, suggesting that the ENTV-2 strain could cause acute infection in goats, which is similar to the acute infection in lambs caused by ENTV-1 [5]. Generally, tumor formation and growth require a certain amount of time. The occurrence of tumors in low-age lambs or goats revealed a possibility of vertical transmission. As the age rose, the number and size of tumors increased in goats (Figure 1). Due to the lack of in vitro cell lines, it is necessary to construct an infectious clone according to the infectious clone strategy of ENTV-1, which may help identify the pathogenicity of the novel ENTV-2 isolate in goats [22]. 

## 4. Conclusions

In summary, we identified the first case of ENTV-2 on a commercial goat farm in Guangdong province of southern China. Our data could help understand the epidemiology and genetic diversity of ENTV-2. The epidemic significance and potential pathogenicity of ENTV-2 in the goat population need to be further investigated.

## Figures and Tables

**Figure 1 viruses-11-00956-f001:**
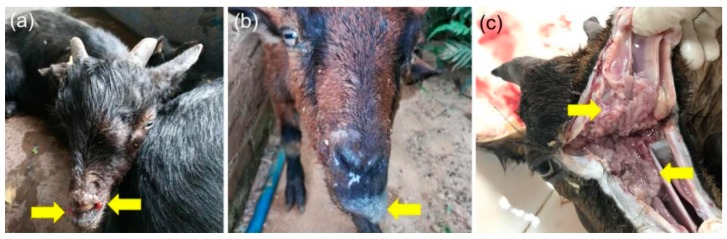
Clinical manifestations and pathological changes of goats affected by enzootic nasal tumor virus (ENTV)-2. (**a**) Bloody polyps appeared around the nostrils of a goat’s nose (Goat 1, about 3-month-old). (**b**) A lot of nasal discharge appeared around the nose (Goat 2, about 10-month-old). (**c**) The nasal cavity (Goat 3, about one-year-old) filled with tumor-like tissues.

**Figure 2 viruses-11-00956-f002:**
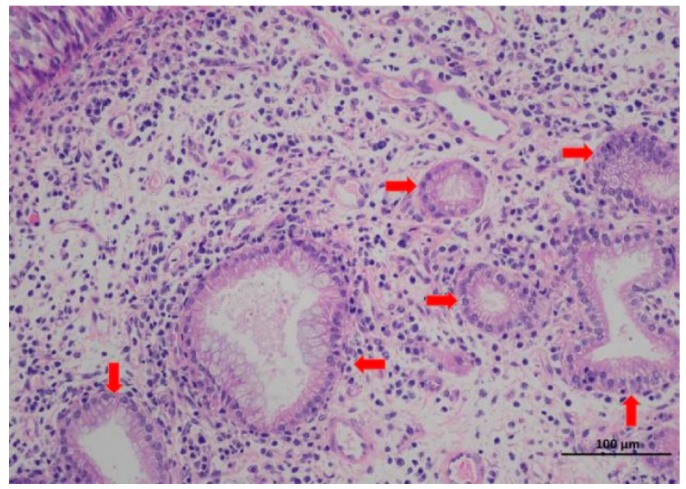
Microscopic feature of tumor tissues in goat 3. There were some different sizes of secretory glandular structures and a lot of interstitial epithelioid cells (labeled by the red arrows) in the figure (Bar = 100 μm).

**Figure 3 viruses-11-00956-f003:**
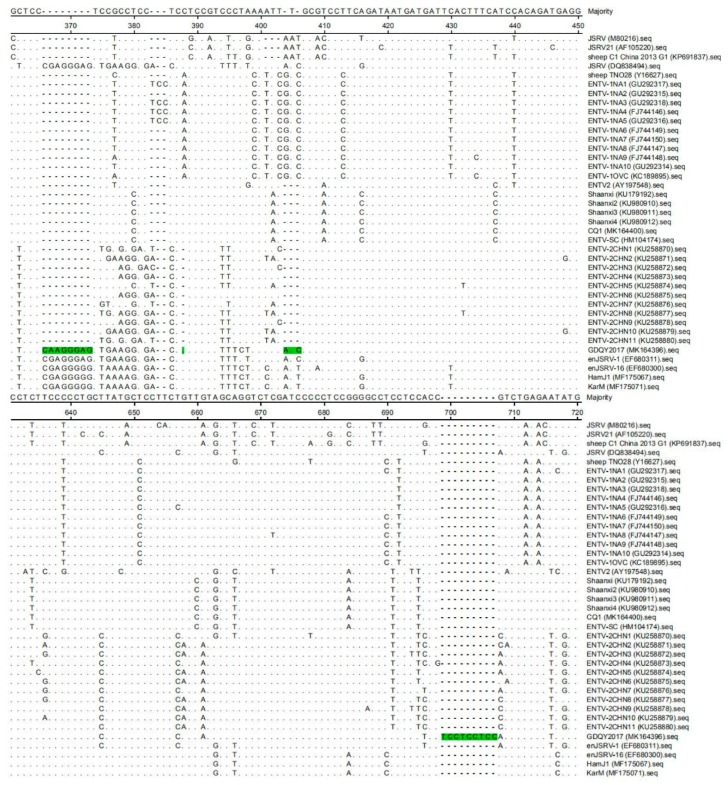
Multiple sequence alignment results of the *gag* gene of ENTV-2 with ENTV-1, jaagsiekte sheep retrovirus (JSRV), and endogenous JSRV (enJSRV). The nucleotide insertion sites in the *gag* gene of GDQY2017 were marked with green color.

**Figure 4 viruses-11-00956-f004:**
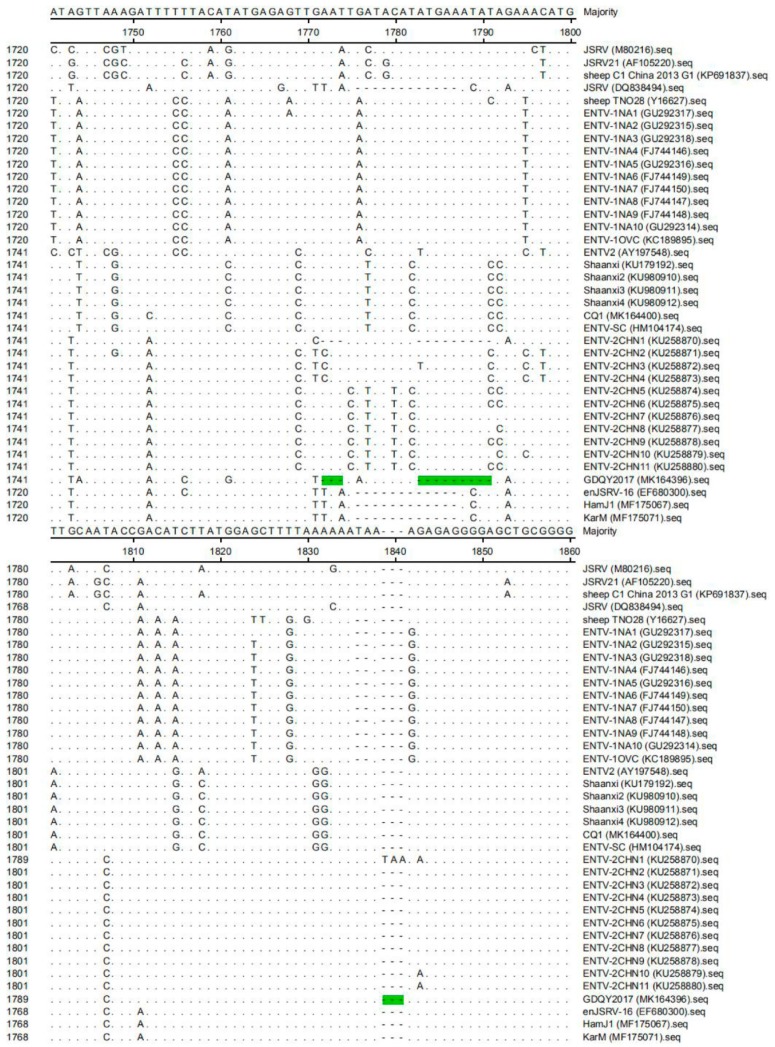
Multiple sequence alignment results of the *env* gene of ENTV-2 with ENTV-1, JSRV, and enJSRV. The nucleotide deletion sites in the *env* gene of GDQY2017 were marked with green color. Moreover, compared to ENTV-2CHN1, GDQY2017 had an additional three-nucleotide deletion (marked with green color) in position 1839–1841 of the *env* gene.

**Figure 5 viruses-11-00956-f005:**
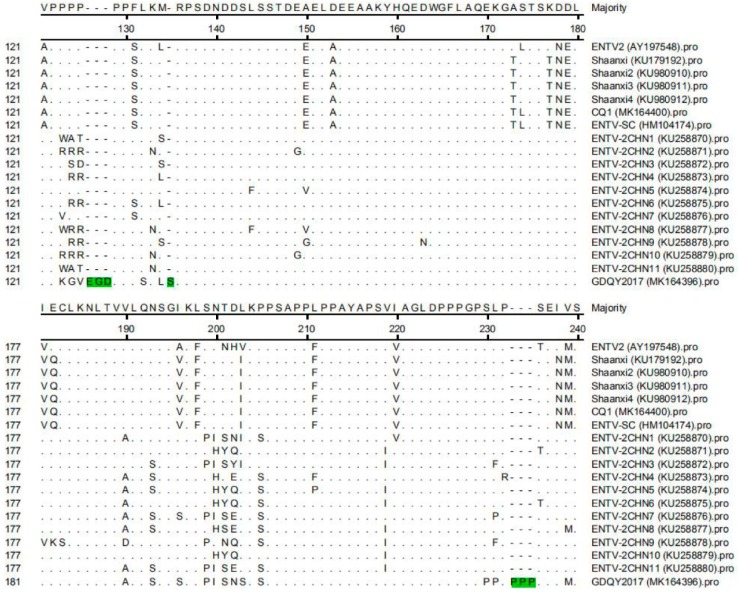
Multiple sequence alignment results of the gag protein of ENTV-2. The amino acid insertion sites (EGD-S-PPP) in the gag protein of GDQY2017 were marked with green color.

**Figure 6 viruses-11-00956-f006:**
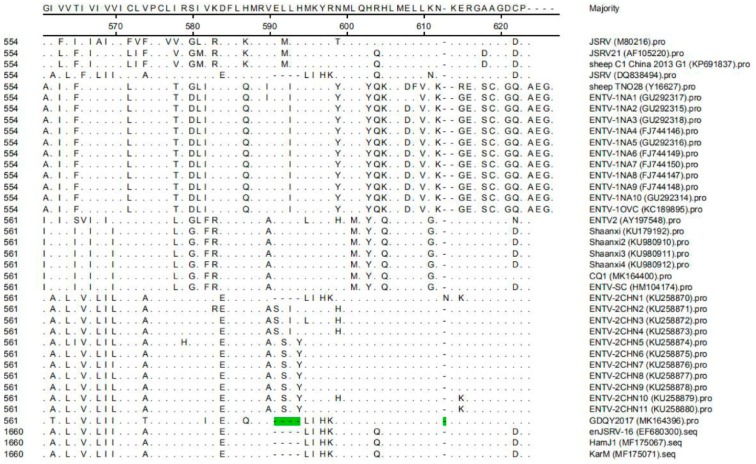
Multiple sequence alignment results of the env protein of ENTV-2 with ENTV-1, JSRV, and enJSRV. The amino acid deletion sites in the env protein of GDQY2017 were marked with green color.

**Figure 7 viruses-11-00956-f007:**
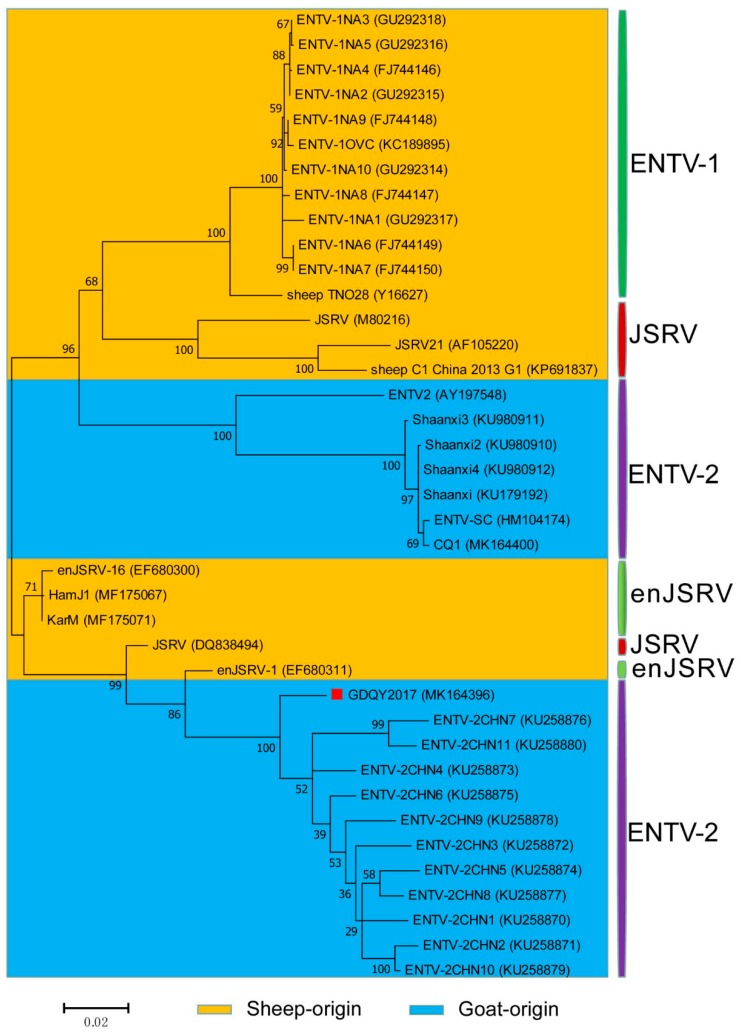
Phylogenetic analysis of the *gag* gene of ENTV-2 with ENTV-1, JSRV, and enJSRV. GDQY2017 described in this study was marked with the red square.

**Figure 8 viruses-11-00956-f008:**
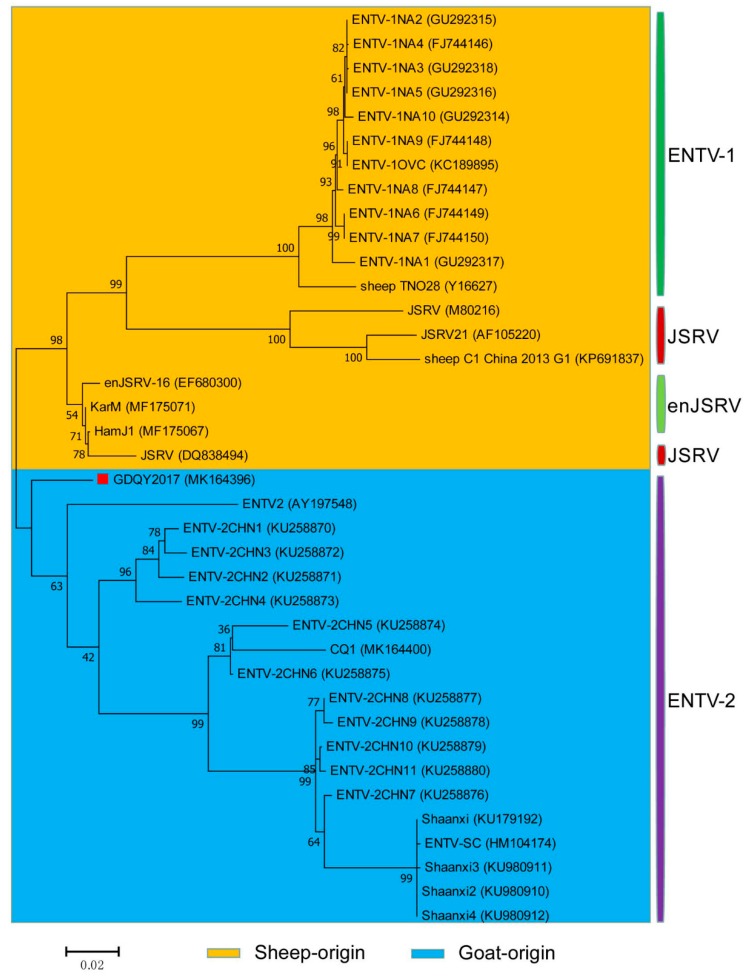
Phylogenetic analysis based on the *env* gene of ENTV-2 with ENTV-1 and JSRV. GDQY2017 described in this study was marked with the red square.

**Table 1 viruses-11-00956-t001:** Primers used in this study.

Primer	Primer Sequence(5′-3′)	Location	Product Length
1F	ACAAGGCATCAGCCATTTTGGTCTGATCCTCTCAACCCCA	1–40	
1R	AGGAGGAGGAGCATCATAACCAGGCTCTGGGTCAGGAATA	627–588	627 bp
2F	GTTTTCCTCGCCACTACTCTTG	151–172	
2R	TACCCAATAAGCGTCGGATGAT	2399–2378	2249 bp
3F	CACTCCTAATTTGTGCCCACG	1848–1869	
3R	GGCCACTGATCGACCCATAC	2934–2915	1087 bp
4F	GAAGAGGTTTGGGGTGTTTTCCCTAGGGACCTCTGATTCTCCTGTGAC	2830–2878	
4R	GTTTAAGACGTTGATGAGCTCGTTCTACAATCCCTTGTCCCTGTGGGT	5087–5040	2258 bp
5F	AGAACGAGCTCATCAACGTCTTAAACATCAACT	5062–5094	
5R	CTTGTTGTTTTATTGTGTCATAGTATATAT	7440–7411	2379 bp

**Table 2 viruses-11-00956-t002:** Similarity (%) of GDQY2017 with other reference retroviruses in goats and sheep.

Isolation (GenBank No.)	Complete Genome	*gag* Gene	gag Protein	*env* Gene	env Protein
JSRV (M80216)	88.2%	86.6%	88.9%	85.9%	89.4%
JSRV21 (AF105220)	88.2%	86.1%	89.4%	86.2%	89.7%
sheep C1 China 2013 G1 (KP691837)	87.6%	85.7%	89.1%	85.8%	89.2%
JSRV (DQ838494)	92.2%	93.6%	87.7%	93.6%	92.2%
sheep TNO28 (Y16627)	88.4%	87.7%	89.7%	87.7%	88.2%
ENTV-1NA1 (GU292317)	88.2%	87.5%	89.4%	87.8%	87.9%
ENTV-1NA2 (GU292315)	88.2%	87.9%	89.4%	87.7%	88.2%
ENTV-1NA3 (GU292318)	88.2%	87.7%	89.3%	87.6%	88.2%
ENTV-1NA4 (FJ744146)	88.2%	87.8%	89.3%	87.6%	88.1%
ENTV-1NA5 (GU292316)	88.2%	87.8%	89.3%	87.7%	88.2%
ENTV-1NA6 (FJ744149)	88.4%	87.9%	89.4%	87.8%	88.1%
ENTV-1NA7 (FJ744150)	88.4%	87.9%	89.4%	87.8%	88.1%
ENTV-1NA8 (FJ744147)	88.3%	87.8%	89.6%	87.8%	88.1%
ENTV-1NA9 (FJ744148)	88.3%	87.8%	89.6%	87.7%	88.2%
ENTV-1NA10 (GU292314)	88.2%	88%	89.6%	87.5%	88.1%
ENTV-1OVC (KC189895)	88.2%	87.7%	89.4%	87.7%	88.2%
ENTV2 (AY197548)	88.2%	87.2%	88.6%	91.2%	89.8%
Shaanxi (KU179192)	91.9%	86.7%	88.7%	89.1%	91%
Shaanxi2 (KU980910)	92%	86.6%	88.7%	89.1%	91%
Shaanxi3 (KU980911)	92.2%	87.2%	88.9%	89.2%	91.1%
Shaanxi4 (KU980912)	91.8%	86.7%	88.7%	89.1%	91%
CQ1 (MK164400)	92.7%	86.6%	88.3%	90.8%	90.5%
ENTV-SC (HM104174)	90.9%	86.5%	88.3%	89%	90.8%
ENTV-2CHN1 (KU258870)	95.7%	96.3%	94.9%	93.3%	92.4%
ENTV-2CHN2 (KU258871)	96%	95.8%	93.3%	93.3%	91.6%
ENTV-2CHN3 (KU258872)	95.8%	95.9%	94.1%	93.5%	92.2%
ENTV-2CHN4 (KU258873)	96.6%	96.8%	94.9%	94.8%	93.7%
ENTV-2CHN5 (KU258874)	95.2%	96%	93.5%	91.7%	91.6%
ENTV-2CHN6 (KU258875)	94.7%	96.8%	95.1%	92.1%	93.1%
ENTV-2CHN7 (KU258876)	94.4%	95.5%	95.1%	90.3%	93.9%
ENTV-2CHN8 (KU258877)	95%	96.6%	94.9%	90.4%	93.5%
ENTV-2CHN9 (KU258878)	92.6%	95.9%	93.3%	90.3%	93.4%
ENTV-2CHN10 (KU258879)	93.3%	96.1%	93.6%	90.1%	93.2%
ENTV-2CHN11 (KU258880)	92.2%	95.6%	95.1%	90.3%	93.4%
enJSRV-1 (EF680311)	94.6%	95.1%	94.5%	-^a^	-^a^
enJSRV-16 (EF680300)	93.3%	91.6%	91.4%	94.6%	94.3%
HamJ1 (MF175067)	93.6%	91.8%	91.7%	94.9%	94.4%
KarM (MF175071)	93.6%	91.8%	91.7%	95%	94.6%

Note: ^a^ It means no comparability between enJSRV-1 and GDQY2017 because the *env* gene of enJSRV-1 was incomplete or uncertain. ENTV, enzootic nasal tumor virus; JSRV, jaagsiekte sheep retrovirus; enJSRV, endogenous JSRV.

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
