# Peer review of "A Novel Enzootic Nasal Tumor Virus Circulating in Goats from Southern China"

_viruses, 2019, doi:10.3390/v11100956_

Round 1

Reviewer 1 Report

The authors clearly improved the manuscript according to the reviewers' comments. 

One question remains about the specificity of the primers. How did the authors excluded the amplification of endogenous retrovirus sequences. did they test their primers on normal goat cells? The primers have strong similarities with JSRV sequences

We suggest to add endogenous JSRV sequences on the trees and alignments

Author Response

Reply to Reviewer 1

Comments and Suggestions for Authors

Moderate English changes required.

Response: We have further revised our manuscript with the help from a native English speaker.

The authors clearly improved the manuscript according to the reviewers' comments. 

One question remains about the specificity of the primers. How did the authors excluded the amplification of endogenous retrovirus sequences. did they test their primers on normal goat cells? The primers have strong similarities with JSRV sequences

Response: We agree with the reviewer and are also very concerned about this important issue. Viral full-length genome sequence was assembled from the sequence information obtained from overlapping five RT-PCR products, ranging from 627 to 2379 bp in size. During the sequence analysis, we have paid great attention to potential non-sense mutations or deletions that might disrupt the open reading frames of the encoded proteins such as Gag or Pol or Env. Despite some insertions or deletions detected in our virus, none of them interfered with the open reading frames of encoded viral proteins. In this regard, we are confident that the presented full-genome sequence information in this study was from an exogenous retrovirus, not from those of endogenous retroviruses living in the genome of sheep or goats.

We suggest to add endogenous JSRV sequences on the trees and alignments.

Response: We added two U.K. endogenous JSRV sequences and two U.S. endogenous JSRV sequences into the phylogenetic trees and sequence alignments.

Reviewer 2 Report

Comments:

(1)Page 3, lines 96-97, show “some different sizes of secretory glandular structures” and “interstitial epithelioid cells” by arrows and/or arrowheads, respectively.

(2)Re: the author’s response to comment 9, thank you for the response that the authors just observed clinical lesions in the nasal cavity. The reviewer suggests that it could be mentioned in e.g. page 9, line 179 after our speculation, as “As far as we checked, any clinical lesions in lungs of the affected goats were not observed (data not shown)”.

(3)English editing would be recommended; e.g. page 9, line 147, spread southward spread.

Author Response

Reply to Reviewer 2

Comments and Suggestions for Authors

Moderate English changes required.

Response: We further revised our manuscript by a native English speaker.

Comments and Suggestions for Authors

(1) Page 3, lines 96-97, show “some different sizes of secretory glandular structures” and “interstitial epithelioid cells” by arrows and/or arrowheads, respectively.

Response: We added the arrows and/or arrowheads in the figure and figure notes.

(2) Re: the author’s response to comment 9, thank you for the response that the authors just observed clinical lesions in the nasal cavity. The reviewer suggests that it could be mentioned in e.g. page 9, line 179 after our speculation, as “As far as we checked, any clinical lesions in lungs of the affected goats were not observed (data not shown)”.

Response: We thank the reviewer’s suggestion. We added it after our speculation.

(3) English editing would be recommended; e.g. page 9, line 147, spread southward spread.

Response: We further revised our manuscript by a native English speaker. We revised some points including the point in the main text.

Round 2

Reviewer 1 Report

The authors adequately responded to the comments except for the exogenous nature of the virus. enJSRV are characterized by open reading frames (unlike other endogenous retroviruses). In their response, the authors stated that "Despite some insertions or deletions detected in our virus, none of them interfered with the open reading frames of encoded viral proteins". Since the complete sequence had been obtained with overlapping segments, they cannot exclude that they sequenced both exogenous and endogenous viruses (the two types of genomes are closely related). Only the full sequence obtained from cloned full genome or large PCR could assure the exogeous nature of the sequenced virus.